# Integrating Explicit and Implicit Fullerene Models into UNRES Force Field for Protein Interaction Studies

**DOI:** 10.3390/molecules29091919

**Published:** 2024-04-23

**Authors:** Natalia H. Rogoża, Magdalena A. Krupa, Pawel Krupa, Adam K. Sieradzan

**Affiliations:** 1Faculty of Chemistry, University of Gdańsk, Fahrenheit Union of Universities in Gdańsk, Bażyńskiego 8, 80-309 Gdańsk, Poland; n.rogoza.927@studms.ug.edu.pl (N.H.R.); krupa.magdalena@outlook.com (M.A.K.); adam.sieradzan@ug.edu.pl (A.K.S.); 2Institute of Physics, Polish Academy of Sciences, Al. Lotnikow 32/46, 02-668 Warsaw, Poland

**Keywords:** molecular dynamics, coarse-graining, force fields, proteins, fullerenes, nanoparticles, nanotoxicity

## Abstract

Fullerenes, particularly C_60_, exhibit unique properties that make them promising candidates for various applications, including drug delivery and nanomedicine. However, their interactions with biomolecules, especially proteins, remain not fully understood. This study implements both explicit and implicit C_60_ models into the UNRES coarse-grained force field, enabling the investigation of fullerene–protein interactions without the need for restraints to stabilize protein structures. The UNRES force field offers computational efficiency, allowing for longer timescale simulations while maintaining accuracy. Five model proteins were studied: FK506 binding protein, HIV-1 protease, intestinal fatty acid binding protein, PCB-binding protein, and hen egg-white lysozyme. Molecular dynamics simulations were performed with and without C_60_ to assess protein stability and investigate the impact of fullerene interactions. Analysis of contact probabilities reveals distinct interaction patterns for each protein. FK506 binding protein (1FKF) shows specific binding sites, while intestinal fatty acid binding protein (1ICN) and uteroglobin (1UTR) exhibit more generalized interactions. The explicit C_60_ model shows good agreement with all-atom simulations in predicting protein flexibility, the position of C_60_ in the binding pocket, and the estimation of effective binding energies. The integration of explicit and implicit C_60_ models into the UNRES force field, coupled with recent advances in coarse-grained modeling and multiscale approaches, provides a powerful framework for investigating protein–nanoparticle interactions at biologically relevant scales without the need to use restraints stabilizing the protein, thus allowing for large conformational changes to occur. These computational tools, in synergy with experimental techniques, can aid in understanding the mechanisms and consequences of nanoparticle–biomolecule interactions, guiding the design of nanomaterials for biomedical applications.

## 1. Introduction

Nanotechnology, an interdisciplinary field combining chemistry, physics, medicine, and engineering, has led to the design and implementation of various nanoparticles in recent decades [1]. Carbon-based nanoparticles, including fullerenes and carbon nanotubes, were among the first discovered and have been widely studied for their unique properties [2,3].

Among carbon nanoparticles, fullerenes are known for their spherical shape, consisting of carbon atoms connected by single and double bonds forming pentagons and hexagons [4]. The most common type is C_60_, also known as buckminsterfullerene, built from 60 sp^2^-hybridized carbon atoms [5]. It has 12 pentagonal and 20 hexagonal sides composing a structure reminiscent of a football [6] with the average diameter of about 0.71 nm [7], calculated using carbon centers of mass. C_60_ naturally occurs in low concentrations [8,9], and can be produced by human activities [10]; however, most C_60_ used today is synthetically produced, with rapidly increasing amounts [11].

C_60_ exhibits unique chemical properties. C_60_ exhibits low water solubility, which can be improved by attaching functional groups [12]. Fullerenes have high electron affinity, allowing efficient electron transport for applications like solar cells [13] and chemical sensors [14,15]. C_60_ derivatives are also used in cosmetics for antioxidative abilities [16] and they are investigated as potential drug delivery systems due to their cage-like structure [17,18].

The increasing environmental presence of C_60_ raises concerns about potential nanotoxicity from skin contact, inhalation, or ingestion [16,19,20]. While the exact fate in the human body is not fully understood [21], research indicates cellular uptake via passive diffusion and endocytosis [22]. As fullerenes enter cells, interactions with proteins should be studied.

### 1.1. Fullerene-Protein Interactions

Interactions of C_60_ and fullerene derivatives (FDs) with certain proteins have been experimentally proven. C_60_ fits well in the HIV-1 protease active site, potentially blocking this key enzyme [23], while monoclonal anti-C_60_ antibodies, including anti-Buckminsterfullerene Fab fragment, were isolated and the Fab fragment with specificity for C_60_ exhibited a competitive inhibition mechanism [24]. Fullerenol, a soluble hydroxylated FD, inhibits various enzymes and proteins [25,26,27,28]. Water-miscible fullerene carboxylic acid also inhibits cysteine and serine proteases [29,30]. Inhibitory effects by FDs have been observed also for several other proteins [31,32,33,34,35]. Docking studies have revealed new potential protein targets for C_60_ binding. Reverse docking identified FKBP and uteroglobin as top candidates [36], while another study found an FD bound to intestinal fatty acid binding protein (IFABP) [37]. Binding to these proteins may alter their activities and biological processes. In vitro experiments confirm potential harmful effects of C_60_ aggregates and fullerenols on tissues, though differences exist between in vitro toxicity and in vivo effects [38,39]. A recent in vivo mouse study found lung tissue changes, increased reactive oxygen species, and decreased ATP production after C_60_ exposure [40].

### 1.2. Theoretical Methods of Studying Nanoparticles

Both experimental and theoretical approaches are used to study nanoparticle properties, each providing valuable observations [41,42]. Experiments allow us to obtain general insight into nanoparticle properties, including nanotoxicity, mentioned above, as well as size, shape, composition, and stability [43]. Theoretical methods, on the other hand, might help us explain underlying mechanisms governing the nanoparticle’s behavior at the molecular and quantum level within a comparatively shorter timescale. For example, in one study, a density functional theory (DFT) method, allowing the exploration of the electronic structure of molecules, was used to precisely determine the exact mode of attaching an analyzed ligand to a gold nanoparticle surface, which was not possible to characterize with experimental methods alone [44]. Another useful theoretical tool in studying nanoparticles is molecular docking, allowing the identification of nanoparticle binding sites on proteins [36]. The molecular dynamics (MD) approach is also used to study potential interactions of nanoparticles with other molecular structures. It depicts the dynamic behavior of a system over time and can be used to examine potential interactions of nanoparticles with biomolecules, for example, cell membranes [45] and proteins [46]. Furthermore, the implementation of coarse-grained models of biomolecules and nanoparticles, such as those used in the MARTINI force field [47,48,49,50], offers a significant advantage in computational efficiency, enabling longer simulations compared to the atomistic approach. However useful, it should be noted that the MARTINI 2 and 3 coarse-grained force fields cannot be effectively used to study large conformational changes of proteins [51,52], as a reference initial atomistic structure is used for generation of backbone and some side-chain parameters [53], and tertiary structure is kept mostly by the elastic network to maintain the structure [54]. This limitation can be somehow alleviated by using the Go-MARTINI variant instead of the elastic network [55].

Therefore, the main goal of this work was to implement explicit and implicit fullerene models into the UNRES force field to enable performing fullerene–protein studies with the ability to study both small and large conformational changes without the necessity to use restraints stabilizing protein structure. On the other hand, using molecular dynamics with a coarse-grained UNRES force field solves the problem of expensive all-atom calculations and allows for 3–4 orders of magnitude longer timescale simulations, while maintaining adequate accuracy.

## 2. Methods

### 2.1. UNRES Model

The UNited RESidue (UNRES) is a highly reduced protein model developed for studies of peptides and proteins. It offers a significant computational advantage, providing a four-order-of-magnitude speed-up compared to traditional all-atom simulations. In the UNRES model [56,57], a polypeptide chain is represented by a sequence of alpha-carbon (C^α^) atoms connected by virtual bonds with united peptide groups (p) placed halfway between consecutive C^α^ atoms, and united side chains (SCs) attached to the C^α^ atoms. United peptide groups and united side chains serve as the primary interaction sites, while the C^α^ atoms are solely responsible for defining the geometry of the polypeptide chain backbone: it is specified by the C^α^–C^α^–C^α^ virtual bond angles θ and C^α^–C^α^–C^α^–C^α^ virtual bond dihedral angles γ. Additionally, α and β are angles defining side-chain center local geometry (Figure 1). The UNRES force field uses a physics-based approach for simulations of protein structure and dynamics. The effective energy function arises from the potential of mean force (PMF) of the system where all degrees of freedom not belonging to the coarse-grained representation have been integrated out. It should be noted that solvent is present in an implicit form and is implemented in the effective energy function. The PMF is then approximated by the generalized Kubo cluster-cumulant series limited to the most important factors. This allows for analytical derivation of the expression for effective energy terms. The effective energy function in the UNRES force field is expressed as follows:(1)$$\begin{array}{cc} & U=w_{SC}\sum_{i<j}U_{SC_{i}SC_{j}}+w_{SCp}\sum_{i\neq j}U_{SC_{i}p_{j}}+w_{pp}^{VDW}\sum_{i<j-1}U_{p_{i}p_{j}}^{VDW}+w_{pp}^{el}f_{2}\left(T\right)\sum_{i<j-1}U_{p_{i}p_{j}}^{el}\\ & +w_{tor}f_{2}\left(T\right)\sum_{i}U_{tor}\left(\gamma_{i},\theta_{i},\theta_{i+1}\right)+w_{b}\sum_{i}U_{b}\left(\theta_{i}\right)+w_{rot}\sum_{i}U_{rot}\left(\theta_{i},\alpha_{SC_{i}},\beta_{SC_{i}}\right)\\ & +w_{bond}\sum_{i}U_{bond}\left(d_{i}\right)+w_{ssbond}\sum_{n_{ss}}U_{ssbond}\left(d_{ss}\right)+w_{corr}^{\left(3\right)}f_{3}\left(T\right)U_{corr}^{\left(3\right)}+w_{turn}^{\left(3\right)}f_{3}\left(T\right)U_{turn}^{\left(3\right)} \end{array}$$
where the *U*s are the energy terms. The function is composed of both long-range and local terms with the addition of multibody components. The long-range components are $U_{SC_{i}SC_{j}}$ (denoting solvent-mediated side-chain–side-chain interaction energies), $U_{SC_{i}p_{j}}$ (corresponding to the excluded-volume potential of side-chain–peptide group interactions), and $U_{p_{i}p_{j}}$ (being peptide group potential split into the Lennard–Jones interaction energy between peptide group centers—$U_{p_{i}p_{j}}^{VDW}$, and the mean electrostatic energy between peptide groups terms—$U_{p_{i}p_{j}}^{el}$). Local components of the polypeptide chain include $U_{tor}$, $U_{b}$, $U_{rot}$, $U_{bond}$, and correspond to the backbone torsional terms, virtual bond angle terms, side-chain rotamer terms, and virtual bond deformation terms, respectively. $U_{ssbond}$ denotes the terms corresponding to disulfide bonds potential [58]. $U_{corr}^{\left(3\right)}$ and $U_{turn}^{\left(3\right)}$ are multibody terms accounting for the coupling of the backbone–local and backbone–electrostatic interactions [59]. Energy terms in the expression are multiplied by appropriate weights ($w_{x}$), which have been reoptimized using maximum likelihood approach [60]. The values associated with factors of order greater than one are scaled by the corresponding temperature coefficients [61], which are defined by $f_{n}\left(T\right)$, where $T_{o}=300$ K. Temperature coefficients are expressed by the following:(2)$$\begin{array}{cc} & f_{n}\left(T\right)=\frac{\ln[\exp(1)+\exp(-1)]}{\ln\left\{\exp\left[{\left(T/T_{o}\right)}^{n-1}\right]+\exp\left[-{\left(T/T_{o}\right)}^{n-1}\right]\right\}} \end{array}$$

UNRES is suitable for the prediction of protein structures and has performed well in biannual CASP experiments [62,63,64,65]. The model was successfully applied in protein folding research [66], free-energy landscapes studies [67], oligomerization of intrinsically disordered proteins [68], and long-time scale simulations of big systems, for example, virus-like particles [69]. An extension to analyze the binding of proteins to carbon nanotubes using an implicit representation of the nanoparticle has also been developed [70] and used to study their impact on model proteins.

**Figure 1 molecules-29-01919-f001:**
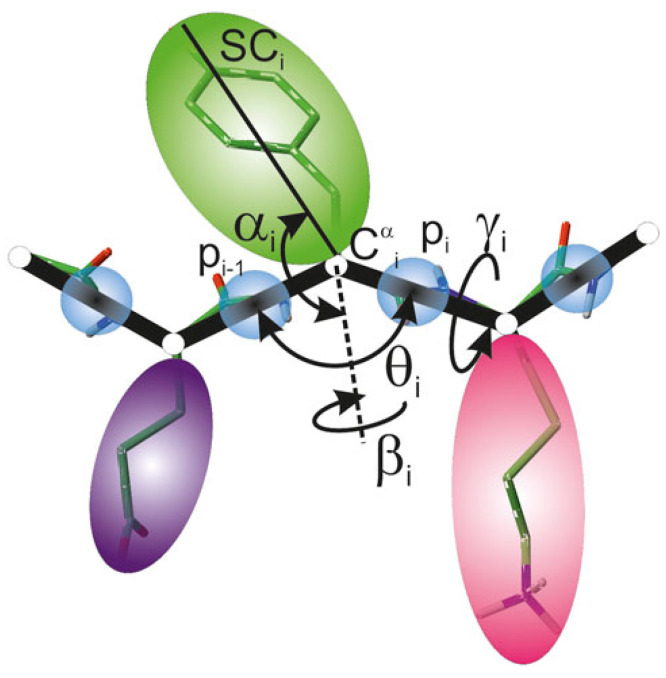
A scheme of polypeptide chain representation in the UNRES model [71]. United peptide groups (*p*) (blue spheres) and united side chains (SC) (spheroids of different colors) attached to C^α^ atoms are the interaction sites. Geometry of the backbone is defined by C^α^–C^α^–C^α^ virtual bond angles θ and C^α^–C^α^–C^α^–C^α^ virtual bond dihedral angles γ. $SC_{i}$ center local geometry is defined by the angle between bisection of $\theta_{i}$ and the $C_{i}^{\alpha }$–$SC_{i}$ vector ($\alpha_{i}$ angle), and the angle of rotation of the $C_{i}^{\alpha }$–$SC_{i}$ vector from $C_{i-1}^{\alpha }$–$C_{i}^{\alpha }$–$C_{i+1}^{\alpha }$ plane ($\beta_{i}$ angle).

### 2.2. C_60_ Model in UNRES Force Field

In this work, we introduced two types of nanospheres to model C_60_ fullerene: implicit and explicit. It should be noted that the implicit model cannot freely move during simulations; however, other molecules, e.g., peptides and proteins, can move around it, while the explicit model allows for full mobility.

#### 2.2.1. Explicit Model

In the explicit model of the C_60_, the fullerene nanoparticle is approximated by 20 alanine side chains (Figure 2B). The alanine side chain was chosen as it is the smallest amino acid that has a side chain with hydrophobic character, similar to C_60_. The number of centers (20) was the maximum number of interaction centers that could be used to approximate C_60_ that remained stable (24 and a larger number of centers were tested, leading to instability of the nanoparticle due to steric clashes). The initial positions of 20 centers are in the form of a regular dodecahedron. It should be noted that no peptide group is present between alanines; however, Cα atoms are present, and face inward toward the nanoparticle and serve as anchors of restraints imposed. Between any two Cα atoms in the fullerene, harmonic restraints were imposed, with the force constant value 10 kcal mol^−1^Å^−2^ to maintain a sphere-like shape. As the C^α^-SC are connected via $U_{bond}$, this allows small distortions of the nanoparticle. As each of the 20 centers is treated separately, there are no $U_{tor}$ or $U_{ang}$ energy terms describing nanoparticle behavior. It should be noted that there is no explicit water–nanoparticle interaction; however, those interactions are implicitly incorporated in the alanine side-chain potentials of mean force. The explicit model of the implemented C_60_ fullerene is based on alanine side chains and interacts with the protein through the Gay–Berne potential (anisotropic modification of the Lennard–Jones potential) (Equation (Equation 3)):(3)$$E_{Gay-Berne}=4\epsilon \left[{\left(\frac{\sigma_{0}}{r-\sigma +\sigma_{0}}\right)}^{12}-{\left(\frac{\sigma_{0}}{r-\sigma +\sigma_{0}}\right)}^{6}\right]$$
where *r* is the distance between the centers of the side chains, σ is the distance corresponding to the zero value of $E_{Gay-Berne}$ for an arbitrary orientation of the particles ($\sigma_{0}$ is the distance corresponding to the zero value of $E_{Gay--Berne}$ for a side-to-side orientation), and ϵ (depending on the relative orientation of the particles) is the van der Waals well depth. For a detailed description, see ref. [72]. The parameters for Equation (Equation 3) are shown in Appendix A.

#### 2.2.2. Implicit Model of Nanoparticle Representation

In the implicit form, the nanoparticle is approximated as a sphere that occupies a given space of a periodic box (Figure 2C). It should be noted that the nanoparticle is immobile and always occupies the same user-defined part of the box; however, other molecules can move around it. The size and shape of the nanosphere are also predetermined and do not change during the course of a simulation; however, it is reasonable in the case of a rigid C_60_. The implicit model of nanosphere allows only to simulate one nanoparticle at a time; however, this representation allows for extremely fast computation. As in case of the explicit nanoparticle, there is no explicit water–nanoparticle interaction; however, those interactions are implicitly incorporated in the protein–nanoparticle potentials of mean force (in σ and ϵ parameters).

The protein-nanoparticle interactions are described by the Kihara potential [73]:(4)$$U_{\mathrm{prot}-\mathrm{nano}}=4\epsilon \left[{\left(\frac{\sigma }{r-R_{0}}\right)}^{12}-{\left(\frac{\sigma }{r-R_{0}}\right)}^{6}\right]$$
where ϵ is the interaction potential well depth, σ is the distance where the interaction potential obtains value 0, *r* is the distance between the protein center of interaction ($p_{i}$ or $SC_{i}$) and the center of the nanosphere, and $R_{0}$ is the size of the nanoparticle. In this particular article, the parameters (ϵ and σ) for interaction with the protein were taken from the phenylalanine side-chain model to approximate the fullerene (Appendix A); however, the application of this nanoparticle model can be extended to various nanoparticles by adjusting the size and the ϵ and σ. The modeled C_60_ had $R_{0}$ set to 3.5 Å.

### 2.3. Protein Selection

To study interactions of proteins with fullerenes, five model systems for which there are suggestions that they can interact with C_60_ fullerene or its derivatives were selected: FK506 binding protein (PDB ID: 1FKF), HIV-1 protease (PDB ID: 1HOS), intestinal fatty acid binding protein (PDB ID: 1ICN), PCB-binding protein (PDB ID: 1UTR), and hen egg-white lysozyme (PDB ID: 1DPX) [36,37,74]. The last was studied in two alternative versions to examine the impact of disulfide residues on the system stability and behavior: with all disulfide bonds present, marked as 1DPX_SS_, and without disulfide bonds, marked as 1DPX_no SS_. It should be noted that dynamic disulfide bond treatment was used, which allows disulfide bonds to break and form during simulations and it is one of the unique features, which can be combined with newly-implemented C_60_ fullerene models [75]. As a reference, a set of UNRES simulations for each of the abovementioned proteins was run without the C_60_ present. These proteins were selected as the previous studies indicated that they can interact with C_60_ fullerene or its derivatives and have relatively small size, which allowed for the comparison with all-atom results, various fold-types and secondary structure contents (1FKF, 1HOS, and 1ICN are mostly formed by β-strands, while 1UTR and 1DPX are mostly helical) and various biological roles played in the organisms. Together, the selected protein set should allow for simple, yet comprehensive, tests of the implemented C_60_ fullerene models.

### 2.4. MD Simulation Details

#### 2.4.1. Coarse-Grained UNRES Simulations

Three different approaches of molecular dynamics (MD) simulations were used in the UNRES force field for each of the selected proteins:Without fullerene nanoparticle (to assess protein stability);With implicit model of fullerene nanoparticle;With explicit model of fullerene nanoparticle.

The starting protein structures were the same as in all-atom Amber simulations after energy minimization. While not strictly required, it was applied for further comparative purposes of coarse-grained and all-atom approaches. Energy minimization of these starting structures in UNRES was carried out with the SUMSL algorithm [76]. All the simulations were conducted with the Berendsen thermostat at 260 K, as the temperature in UNRES is not yet fully optimized and 260 K is recommended [77]. The number of steps in each simulation was set to 4,000,000, which, with a time step of 4.89 fs and 1000 times speed-up of the UNRES force field, resulted in almost ∼20 μs laboratory time long simulations. A total of 20 trajectories were run for each type of approach for each protein, totaling about ∼400 μs for each system variant.

#### 2.4.2. All-Atom Amber Simulations

For comparison, we conducted individual 1000 ns molecular dynamics (MD) simulations on selected proteins with and without a C_60_ fullerene particle using Amber22 [78] with GPU implementation. AnteChamber was employed to obtain the C_60_ molecule parameters, taking advantage of the carbon-only molecule’s rigid nature, eliminating the need for sophisticated tools for charge estimation or conformational optimization. Bond and angle parameters were derived from the ff19sb [79] protein and GAFF [80] general force fields. The manual placement of C_60_ in protein binding pockets was performed using PyMol. These conformations were then surrounded by water molecules modeled using the four-point OPC water model [81]. The system was configured as a truncated octahedron periodic boundary box with a minimum water layer thickness of 15 Å. To neutralize the charge, Na^+^ or Cl^−^ ions were added, and the system underwent parameter and topology file generation using tLeap, part of AmberTools23 [82]. Solvated proteins underwent energy minimization (10,000 steps in total, utilizing 4000 and 6000 steps with steepest descent and conjugate gradient algorithms, respectively), followed by heating and equilibration in the NPT ensemble over 1 ns with a time step of 1 fs. Subsequently, 1000 ns conventional MD simulations with a time step of 2 fs were conducted in the NVT ensemble with GPU-compatible calculations, saving snapshots every 100 ps. Nonbonded interactions were treated with the PME method, and the cutoff value was set to 9 Å while the temperature of 300 K was maintained by the Langevin thermostat. A parallel set of simulations without the C_60_ fullerene was also performed. To investigate the impact of disulfide residues on system dynamics and stability, lysozyme was simulated in two variants: one with all disulfide bonds present and another with all disulfide bonds absent.

### 2.5. Analysis

#### 2.5.1. Binding Energy Estimation

As a part of the study, binding energies between the protein and C_60_ fullerene were estimated. In the implicit model of fullerene in UNRES, the binding energy results directly from the interaction potential, while in the explicit model, the energy of the complex ($E_{\mathrm{complex}}$) was obtained directly from the simulation. Meanwhile, the energies of the protein ($E_{\mathrm{protein}}$) and the C_60_ nanoparticle ($E_{\mathrm{nano}}$) in bulk water (implicit solvent) were calculated based on the same conformations as in the complex. The final binding energy was then calculated as
(5)$$E_{\mathrm{binding}}=E_{\mathrm{complex}}-\left(E_{\mathrm{protein}}+E_{\mathrm{nano}}\right)$$

For all-atom simulations, the Molecular Mechanics with Generalized Born and Surface Area Solvation (MM-GBSA) method was used to estimate binding energies between selected proteins and C_60_ fullerene using 100 snapshots evenly distributed from the second halves of the trajectories [83]. This procedure is analogous to the one performed for the explicit C_60_ model, with a single difference: as the all-atom simulations were run in an explicit solvent model, the conversion to an implicit solvent in MM-GBSA calculations adds solvation entropy to the results, which is not present for coarse-grained UNRES simulations as UNRES employs an implicit solvent in both MD simulations and analysis. Therefore, the obtained enthalpy of binding ΔH is often called the effective binding energy. Additionally, the normal-mode analysis of harmonic frequencies method was used to predict the entropic contribution based on five snapshots per system due to the high computational cost of the method, providing Gibbs free energy:(6)ΔG=ΔH−TΔS
where ΔG is the binding free energy, ΔH is the enthalpy of binding or effective binding energy, *T* is temperature, and ΔS is the entropy change of the system upon complex formation.

#### 2.5.2. All-Atom Structure Reconstruction

In order to further compare all-atom results with coarse-grained ones, the UNRES protein trajectories were first converted into all-atom representation using the PULCHRA tool [84]. Missing hydrogen atoms were added with the tLeap program. In the case of C_60_ in explicit simulations, the multi-alanine representation was replaced with an all-atom equilibrated one, based on the geometrical centers of these. For the implicit fullerene model, the all-atom C_60_ structure was inserted in the place of an approximated nanoparticle, based on the geometrical centers of the all-atom structure and the immobile nanosphere.

#### 2.5.3. CPPTRAJ Analysis

To evaluate properties of proteins and C_60_ during simulations, an analysis with the CPPTRAJ program [85] from AmberTools23 [82] was conducted for both all-atom and reconstructed UNRES trajectories. It included calculations for root mean square deviation (RMSD) for Cα atoms, mass-weighted root mean square fluctuation (RMSF) for Cα atoms, and radius of gyration with maximum radius of gyration, both for Cα atoms. Moreover, the solvent-accessible surface area (SASA) values for all atoms were calculated for protein structures (in all simulations), as well as for protein–fullerene complexes and for single C_60_ molecules (in simulations with C_60_). Additionally, native contacts analysis was run. The contact cut-off distance was set to 8 Å. If a given C_60_ interaction center was within an established cut-off distance to any protein atom, it was considered to be in contact. In addition, average numbers of residues in given secondary structure were obtained.

#### 2.5.4. Contact Probability

To analyze the stability of C_60_ binding to proteins over time, the evaluation of contact probabilities of given protein residues with explicitly defined fullerene nanoparticle during MD simulations in the UNRES force field was conducted. It was calculated from two perspectives: that of the fullerene and that of the protein, with a contact cut-off distance between residues and centers of interactions of 8 Å. Each final contact probability value was then obtained as an average from all 20 trajectories.

## 3. Results

### 3.1. Stability of a Nanoparticle in a Complex

As the selected proteins are known to interact with fullerenes and FDs, we first analyzed the stability of the C_60_ fullerene in the protein binding pockets. Appendix A shows that high contact probabilities (close to or equal to 1) were observed for all 20 nanoparticle interaction centers in 1HOS, 1ICN, and 1UTR. Additionally, for 1FKF, an interesting contact pattern can be observed. Smaller values (0.5–1) are visible along the time axis, but at no stage is dissociation observed (at least one center from a nanoparticle has contact probability 1). This may be explained by a consistent rotation of the protein around a C_60_ molecule buried in a shallow binding pocket. Low contact probabilities are visible for the 1DPX protein (1DPX_no SS_, 1DPX_SS_), where dissociation of the protein–nanoparticle complex was observed in some trajectories.

Plots of contact probabilities from the protein residues’ point of view (Appendix A) reveal nonspecific binding of a nanoparticle to the 1DPX protein (Appendix A). For the rest of the proteins, the interaction pattern is more specific.

For the FKBP protein (1FKF) (Appendix A), four residues maintained constant contact with the C_60_ nanoparticle during each of the 20 MD trajectories: Tyr26, Phe46, Leu74, and Phe99. Some protein residues formed contact during the course of the simulation, which generally remained stable until the end (Val4, Val24, Met49, Pro88, Gly89, Val101). In contrast, several residues lost their initial contact with C_60_ (Phe36, Ile56, Trp59, Ile76, Tyr82, His87, Ile90). The constant contact results for Tyr26, Phe46, and Phe99 agree with experimental findings. An X-ray crystallography study of the FKBP-FK506 complex revealed a shallow binding cavity, with the side chains of Tyr26, Phe46, Phe99, Val55, and Ile56 building the sides of the hydrophobic pocket [86]. Based on these results, together with MD trajectory visualization analysis in PyMOL, C_60_ indeed remained within the binding pocket of FKBP throughout all simulations. All these observations are in agreement with the results obtained from all-atom trajectories.

The HIV-1 protease (1HOS) is a homodimeric enzyme; hence, contact probabilities for monomers A and B were plotted separately (Appendix A). Similar interaction patterns are noticeable for both chains. For each monomer, three large protein fragments with significant contact probability values are visible. They correspond to specific parts of the protein that keep the C_60_ molecule in the enzyme’s binding site. Starting from the beginning of the protein sequence, in the first group, the residues with the most probable contacts with C_60_ throughout all simulations were Asp25, Ala28, and Val32. In the second segment, residues Ile47-Gly49, Ile54, and Val56 maintained constant contact with C_60_ during the majority of simulations; however, for Ile54 and Val56, the contacts were more probable for chain B. The third section shows Leu76-Thr80 and Ile84 residues as the most probable, with Gly78-Thr80 being more probable in chain B. The experimental active site of the HIV-1 protease consists of residues Asp25-Gly27, which in this analysis corresponds to the first group. Another important part of the HIV-1 protease is the so-called flaps. They shield the enzyme’s active site, built from residues Lys43 to Gln58 [87], belonging to the second group here. Moreover, residues Val77-Val82, belonging to the third group in this analysis, were associated with conformational changes causing narrowing of the active site cavity [88]. Taking all findings into account, the C_60_ nanoparticle stayed in the HIV-1 protease’s active site during course of MD simulations.

For the IFABP protein (1ICN), particular residues building an internal channel identified as the binding site in one study [89] were mostly the ones that maintained contact with the C_60_ nanoparticle in our simulations: Trp6, Val8, Tyr14, Phe17, Met18, Met21, Ile23, Leu36, Leu38, Val49, Lys50, Glu51, Ile58, Val60, Phe62, Phe68, Tyr70, Trp82, Leu89, Gly91, Phe93, Ala104, Val105, Gln106, Gln115, Thr116, Tyr117, Arg126, and Phe128 (Appendix A).

Similarly to 1HOS, probabilities for both monomers of uteroglobin (1UTR) were plotted separately (Appendix A). The same residues in both chains remained in constant contact with a nanoparticle: Phe5, Leu9, Leu12, Leu13, Leu40, Leu43, Val44, Leu47, Ile55, and Thr59. This agrees with experimental results, where the same residues created active sites for polychlorinated biphenyl molecule [90].

In the case of lysozyme (1DPX), the number of interactions between the protein and the explicit C_60_ fullerene model is significantly lower than for the other studied proteins. This trend agrees with the all-atom simulations, in which this protein interacts in the least strong manner. However, this observation does not agree with implicit C_60_ simulations, in which it interacted strongly in the majority of the simulations. The interactions are mainly maintained by residues 60–63 for reduced lysozyme and 60–63 along with 106–111 fragment when disulfide bonds are present.

In general, results for explicit C_60_ representation are in agreement with experimental findings on potential binding of fullerene to the studied proteins [36,37].

### 3.2. Strength of the Protein–Nanoparticle Interaction

Estimated binding energies (Table 1) were different for protein–C_60_ complexes in each of the approaches. Relatively lower values were observed for the implicit representation, followed by the all-atom representation, and the highest values were obtained with the explicit model. A correlation trend is visible in the data—there is a strong linear correlation between energy (R^2^ = 0.97) values from all-atom and UNRES explicit C_60_ simulations (Figure 3A). Hence, all binding energies from all-atom simulations of protein binding to C_60_ are in good agreement with energies obtained from simulations with the explicit representation of the nanoparticle. It should be noted that in the case of the ${\mathrm{1DPX}}_{SS}$ simulation, the average interaction energy is positive, indicating repulsion between the nanoparticle and lysozyme. This suggests that the nanoparticle is only kinetically bound to the lysozyme and cannot overcome the dissociation barrier in certain trajectories.

Similar observations are not applicable to the implicit approach, particularly because of the strong binding of C_60_ to the 1DPX protein (both 1DPX_no SS_ and 1DPX_SS_), which was not observed in the other methods. This leads to a lowering of correlation (R^2^ = 0.44); if the 1DPX protein is excluded, a high correlation is observed (R^2^ = 0.95). This indicates that the implicit model generally gives good binding energies but in some cases can overestimate the binding strength.

Even higher correlation is observed when the entropy contribution is included in the binding energy in the all-atom force field (Figure 3B). The correlation coefficients are R^2^ = 0.98 and R^2^ = 0.52 for the explicit and implicit models of the nanoparticle, respectively. This is quite understandable, as in the UNRES force field, the energy function is based on PMF, which contains an entropy contribution from averaging out the omitted degrees of freedom, also called restricted free energy. This also indicates that entropy plays an important role in nanoparticle binding.

The strength of binding is connected to the number of C_60_ interactions with protein residues over time for explicit simulations (Appendix A). All nanoparticle centers were in constant contact with protein residues for 1ICN and 1UTR (Appendix A), which were the proteins with the strongest binding. A slightly weaker interaction was identified for the 1HOS protein, where the majority of the contact was preserved during simulations (Appendix A). Similarly, smaller contact probability values correspond to higher energy observed for 1FKF (Appendix A) and 1DPX (Appendix A).

### 3.3. Nanoparticle Impact on Protein Structure

Analysis of the RMSD shows that the presence of the C_60_ fullerene does not significantly destabilize any of the analyzed proteins in any of the simulation schemes (Figure 4). Some small destabilizations upon C_60_ binding may be observed for 1KFK and 1ICN in all-atom trajectories; however, these differences are negligible and are probably attributed to normal fluctuations of the protein chain at room temperature. However, in the case of all-atom simulations of 1DPX without the disulfide bonds present, a strong opposite effect is observed, as the presence of C_60_ fullerene stabilizes the protein structure. It should be noted, though, that this effect may simply be a result of slowing down the partial unfolding process occurring in the absence of disulfide bonds in the all-atom force field and would be observed if a sufficient timescale of simulations were reached, as most proteins are known to be semistable in the absence of some, or even all, disulfide bonds. This is further confirmed by the lack of the C_60_ effect on the stability of lysozyme in coarse-grained simulations. It should be noted that higher RMSD values are observed for the coarse-grained simulations than all-atom, which is caused by the simplified representation of proteins in the UNRES model, longer computational timescales, averaging over 20 trajectories, and inaccuracies in the reconstruction to all-atom representations. However, UNRES is known for accurately capturing protein behavior and ligand binding [91,92], despite the low resolution of the structures.

When the influence of the nanoparticle on individual residue is analyzed (RMSF plots; Figure 5) the UNRES force field in general reveals larger fluctuations than observed in all-atom, probably as the longer simulations and, therefore, larger conformation changes could be observed. The largest differences are observed for the C-terminal fragment during all-atom 1DPX simulations without disulfide bonds; however, those changes are not confirmed by UNRES simulations. In the case of 1FKF, the largest difference is observed for region 55–65 (Figure 5A). This region is in contact with fullerene in some trajectories while losing contact in others, which might lead to conformation diversity in this region. In the 1HOS protein (Figure 5B), residues 45–55 are stabilized by fullerene, while in the UNRES force field, this region was destabilized on average by C_60_; however, in many trajectories, stabilization could be observed as it has an extremely wide distribution in that region. For the 1ICN protein (Figure 5C) in both all-atom and UNRES simulation, region 53–58 is destabilized by the nanoparticle. In the case of the 1UTR, no significant influence of the nanoparticle on fluctuation could be observed.

Based on the average number of residues in a given secondary structure among all simulations with different approaches (Figure 6), we see that for all proteins, alpha and beta secondary structures remained, which, together with the lack of significant changes in observed radii of gyration (Figure 7), indicates that nanoparticle binding did not denature the protein. All-atom results for all proteins are mostly in agreement with the initial distribution of secondary structure; however, the UNRES force field tends to slightly distort secondary structure, which is visible for the 1HOS protein in Figure 6B, where the number of residues building the beta-sheet structure decreased. This effect is also attributed to the imperfection of the all-atom reconstruction and the way the DSSP algorithm predicts secondary structure elements based mostly on the hydrogen bond network, which is especially prone to inaccuracies during all-atom structure reconstruction.

Analysis of SASA values shows that the UNRES reconstructed model possesses constant but 30% higher values than all-atom models (Figure 8). This effect is attributed to the imperfection of the all-atom reconstruction and the tendency of the UNRES coarse-grained force field to make proteins more soluble. Interestingly, there is about 10% higher SASA for protein when a nanoparticle is present, indicating that the nanoparticle prevents the collapse of hydrophobic residues that are in contact with that nanoparticle. Moreover, in most cases, the addition of the nanoparticle does not increase overall SASA. Interestingly, we found no correlation between SASA increase due to the presence of the nanoparticle and the number of contacts (Figure 9). However, there is a visible relation between the number of contacts and the strength of interaction, as the weakest interaction is lysozyme (1DPX), followed by the 1FKF protein, and at the same time, 1DPX reveals the lowest number of contacts followed by 1FKF. It should be noted that the more generalized interaction site, where plenty of hydrophobic contacts can be formed, exhibits stronger binding affinity (1ICN and 1UTR) than the specific binding pocket, in which both hydrophobic and electrostatic interactions are normally formed with the ligand (1FKF and 1DPX). Therefore, it seems that the FDs would exhibit stronger binding affinity to binding pockets than C_60_; however, they may still lack elasticity to form strong interactions within well-defined binding pockets. The largest difference in the number of contacts between all-atom and coarse-grained simulations is observed for the 1HOS protein; this difference arises from the depth of penetration of the nanoparticle. In all-atom simulations, the nanoparticle is deeply buried, while in the UNRES simulation it is surface-bound (Figure 10). The low number of contacts for 1DPX is also visible in the representative structure, where the nanoparticle is barely bound. It should be noted that in some cases (1ICN and 1UTR), the nanoparticle can penetrate deeply into the protein structure, similarily to the all-atom simulations.

The presence of the C_60_ fullerene has a very limited impact on disulfide bond stability during MD simulations (Figure 11), both in simulations with implicit and explicit C_60_ models. The only small difference can be seen in the presence of the implicit C_60_ model, which destabilizes the least stable disulfide bond, Cys64-Cys80. In general, all of the disulfide bonds are stable during simulations, keeping the respective parts of the protein close to each other. In simulations without the disulfide bonds, these parts tend to slightly rearrange, not resulting in a significant increase in the RMSD, but, nevertheless, increasing the distance between the sulfur atoms to about 10 Å for Cys30-Cys115, Cys64-Cys80, and Cys76-Cys94, and 15 Å for Cys6-Cys127. The largest distance for the latter is caused by the fact that the cysteine residues involved in this disulfide bond are placed in the most distant parts of the protein. Interestingly, a nanoparticle, when simulated in an coarse-grained explicit form, seems to have a small but noticeable stabilizing impact on the region in which Cys6 and Cys127 are present, when disulfide bonds are absent. A similar effect is observed in all-atom simulations without disulfide bonds—the presence of the C_60_ fullerene stabilizes the C-terminal part of the protein (Figure 5), which is normally rigidified by the Cys6-Cys127 disulfide bond. It should be noted that due to the imperfections of the all-atom reconstruction, even with the presence of the disulfide bond, the distance between sulfur atoms is equal to about 4 Å, instead of the typical all-atom distance of 2.05 Å.

## 4. Discussion

The results presented in this study demonstrate the successful implementation of both explicit and implicit C_60_ fullerene models into the UNRES coarse-grained force field for molecular dynamics simulations of protein–nanoparticle interactions. The explicit model, representing C_60_ as 20 alanine side chains in a dodecahedral arrangement, showed good agreement with all-atom simulations in terms of predicting protein flexibility, the position of the fullerene in protein binding pockets, and estimated binding energies [36,37,74]. The implicit model, treating C_60_ as an immobile sphere interacting with the protein via a Kihara potential, provided a computationally efficient alternative, although it tended to overestimate binding strength in some cases compared to the explicit model and all-atom simulations [36]. This overestimation of binding strength by the implicit C_60_ model may be attributed to its inherent limitations and simplifications. The lack of flexibility may prevent the nanoparticle from adapting to the protein surface and forming more realistic interactions, potentially leading to an overestimation of binding strength. Furthermore, the implicit model relies on predetermined parameters, such as the interaction potential well depth (ϵ) and distance (σ), which were taken from the phenylalanine side-chain model to approximate C_60_. These fixed parameters may not be optimal for all protein systems, resulting in inaccuracies in binding energy calculations. To address these limitations, future work will focus on optimizing these parameters to better fit the binding affinities predicted by all-atom methods. This optimization is expected to improve the accuracy of the implicit C_60_ model across a wider range of protein–nanoparticle systems. Additionally, there are ongoing improvements to the UNRES potentials, parameters, and method for reconstructing all-atom models from coarse-grained representations optimized on UNRES models rather than experimental conformations. These advancements are expected to enhance the accuracy of local protein structure predictions.

In this study, all analyses are based on unrestrained UNRES simulations, in which the investigated systems are free to change conformations during trajectories, which allows for a more extensive exploration of the protein conformational space compared to models that rely on restraints or structure-based potentials. By allowing the system to evolve freely during simulations, UNRES can capture a wider range of conformational states and transitions that may be relevant to protein function and dynamics. This approach differs significantly from other popular coarse-grained models, such as MARTINI, which not only applies restraints (elastic network or Go-like potentials) to maintain protein structural stability but also utilizes potentials based on the reference (initial) structure, further biasing possible protein conformational changes and structural stability [51,53]. However, it should be noted that the use of a coarse-grained representation and unbiased, unrestrained trajectories may, in rare cases, lead to an overdestabilization of proteins in some trajectories. Therefore, average properties, rather than single outlying events, should be analyzed to obtain a more accurate understanding of the system’s behavior.

Analysis of contact probabilities between C_60_ and protein residues revealed distinct interaction patterns for the different proteins studied. For example, specific binding sites were observed for FK506 binding protein (1FKF), while more generalized interactions were seen for intestinal fatty acid binding protein (1ICN) and uteroglobin (1UTR) [36]. These findings are consistent with previous experimental and computational studies identifying potential C_60_ binding sites on these proteins [23,24,89].

Importantly, the presence of C_60_ did not significantly destabilize any of the proteins in either the explicit or implicit simulations, as evidenced by analysis of RMSD, radius of gyration, and secondary structure [36]. This suggests that C_60_ binding does not induce major conformational changes or denaturation, in agreement with experimental observations [21,38]. However, subtle changes were noted in some cases, such as the slight destabilization of 1FKF and 1ICN in all-atom simulations upon C_60_ binding [36].

The coarse-grained UNRES simulations enabled access to much longer timescales (up to 10 μs) compared to all-atom MD (1 μs), while still capturing key aspects of protein–C_60_ interactions [36]. More importantly, the real-time calculation of 1 μs takes less than an hour on a single CPU core of a standard PC with the UNRES model, while all-atom MD simulations for the investigated systems required 3–5 days on a modern GPU. This time would increase by more than tenfold if a state-of-the-art CPU node with 128 cores were used. This means that with the same computational resources, not only are UNRES coarse-grained simulations about three orders of magnitude longer, but also, two orders of magnitude more trajectories can be run. It should be noted that due to the recent implementation of GPU support in the UNRES package [93], it can now be utilized for simulations. However, owing to its simplified coarse-grained representation, it would primarily offer a significant speed-up for very large systems. This highlights the utility of multiscale approaches leveraging coarse-graining to investigate nanoparticle–biomolecule interactions at experimentally relevant scales that are challenging for conventional all-atom MD [41,44].

The development of accurate coarse-grained force fields is crucial for reliable simulations of protein–nanoparticle interactions. Improvements in parameterization methods, such as maximum likelihood optimization [60], have led to more transferable and robust coarse-grained models. Furthermore, the integration of machine learning techniques, such as graph convolutional networks [94] and diffusion models [95], with coarse-grained simulations has the potential to further enhance the accuracy and efficiency of these methods. Other coarse-grained models have also been successfully applied to study the interactions of fullerenes and their derivatives with lipid membranes [45]. For example, Nisoh et al. used the MARTINI force field to investigate the effects of fullerenes on plasma membrane properties, revealing distinct interaction patterns and potential mechanisms of cellular uptake [45]. It should be noted that our C_60_ model can also be extended through minor modifications, such as altering single or multiple alanine side chains to represent other amino acid residues, like serine in an explicit model or through modifications of σ and ϵ parameters in an implicit model. This adaptation enables the C_60_ model to encompass various fullerene derivatives (FDs), such as fullerenols, thereby greatly expanding the range of simulated systems.

In conclusion, the integration of explicit and implicit C_60_ models into the UNRES force field, coupled with recent advances in coarse-grained modeling and multiscale approaches, provides a powerful framework for investigating protein–nanoparticle interactions at biologically relevant scales. These computational tools, in synergy with experimental techniques, can aid in understanding the mechanisms and consequences of nanoparticle–biomolecule interactions, guiding the design of nanomaterials for biomedical applications. Future work should focus on extending these models to other fullerene derivatives and biomolecular systems, as well as incorporating advanced sampling techniques and machine learning methods to further enhance the accuracy and efficiency of coarse-grained simulations.

## 5. Conclusions

In this work, explicit and implicit C_60_ fullerene models were successfully integrated into the UNRES coarse-grained force field and applied to study interactions with several proteins without using any restraints, thus allowing for large conformational changes. The explicit model, parameterized based on all-atom simulations, yielded results consistent with all-atom MD in terms of protein flexibility, C_60_ binding poses, and interaction energies. The implicit model provided a more computationally efficient alternative, although it sometimes overestimated binding strength.

Analysis of the simulations revealed protein-specific interaction patterns, with some exhibiting localized C_60_ binding sites and others more generalized interactions. Importantly, C_60_ did not significantly disrupt protein stability in most cases, and even led to an increase in the stability of lysozyme when no disulfide bonds were present. The coarse-grained simulations accessed submillisecond timescales, enabling observation of events beyond the reach of conventional all-atom MD.

Overall, this study demonstrates the utility of coarse-grained MD, specifically the UNRES force field, for investigating protein–nanoparticle interactions. The multiscale approach of integrating insights from all-atom simulations into coarse-grained models can provide a powerful framework for probing these systems at experimentally and physiologically relevant scales. The developed C_60_ models can be readily extended to other carbon nanoparticles, such as fullerene derivatives, and combined with advanced sampling techniques to further explore the mechanisms and consequences of nanoparticle–protein interactions. These computational tools, in synergy with experiments, can aid in understanding the biological effects of nanomaterials and guide the design of nanoparticles for biomedical applications. 

## Figures and Tables

**Figure 2 molecules-29-01919-f002:**
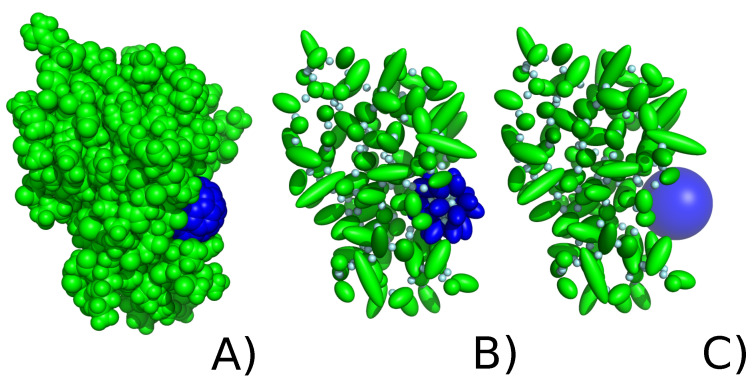
FK506 binding protein (green) with C_60_ fullerene (blue) in (**A**) all-atom representation, (**B**) UNRES coarse-grained protein representation with explicit coarse-grained fullerene representation, (**C**) UNRES coarse-grained protein representation with implicit coarse-grained fullerene.

**Figure 3 molecules-29-01919-f003:**
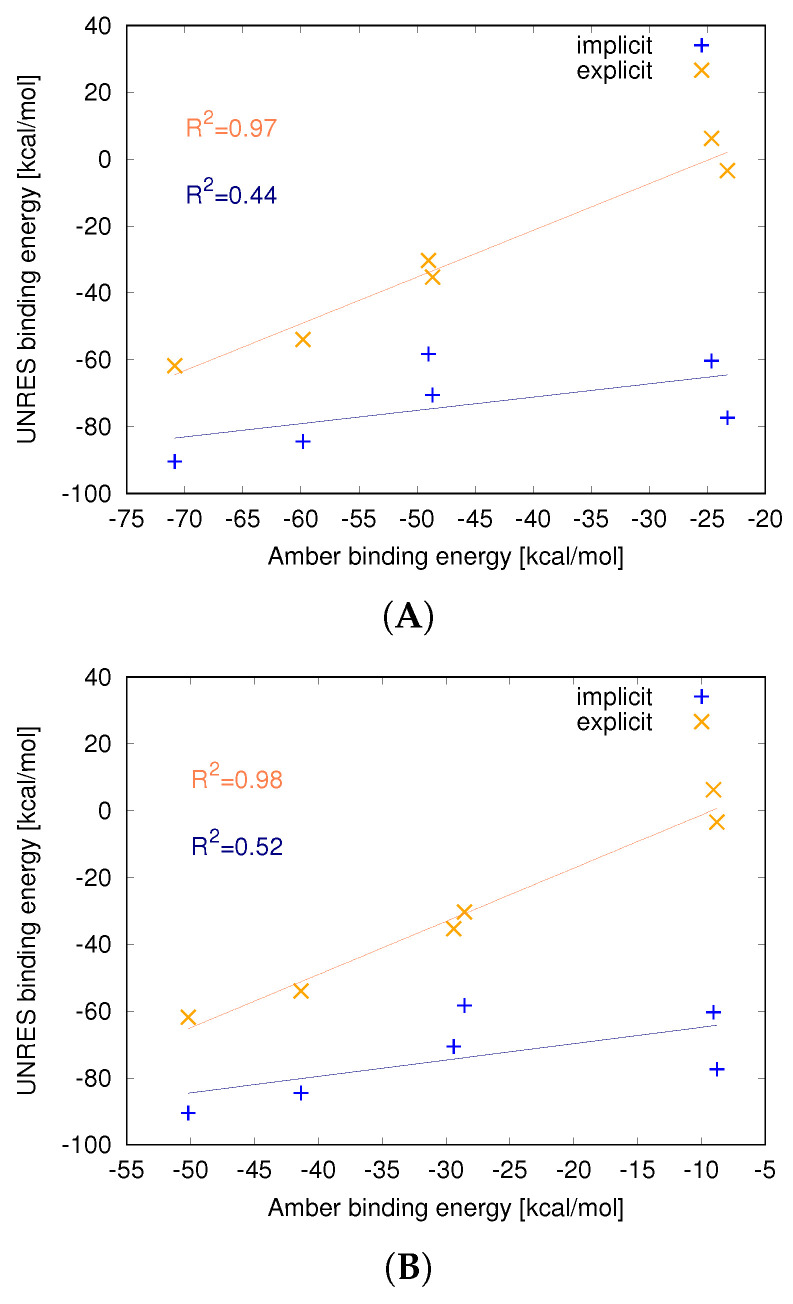
Correlation between average energy of interaction between C_60_ fullerene in implicit (blue) or explicit (orange) representations in the UNRES coarse-grained force field and ΔH (**A**) and ΔG (**B**) in the all-atom representation in the Amber force field.

**Figure 4 molecules-29-01919-f004:**
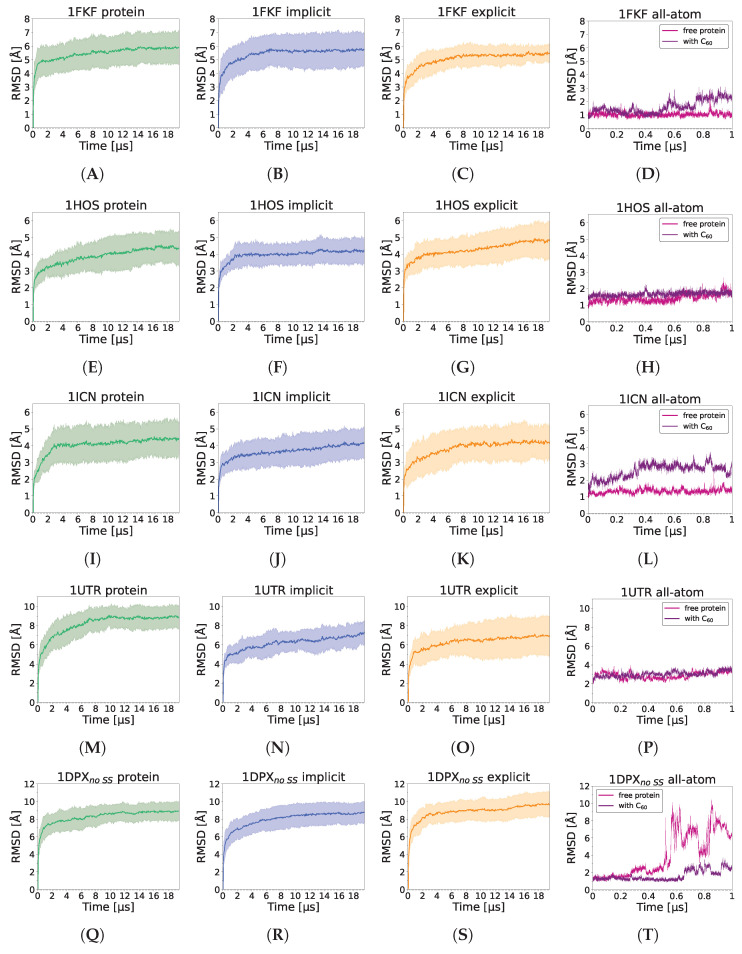
RMSD values for 1FKF (**A**–**D**), 1HOS (**E**–**H**), 1ICN (**I**–**L**), 1UTR (**M**–**P**), 1DPX_no SS_ (**Q**–**T**), and 1DPX_SS_ (**U**–**X**) protein performed with UNRES force field in simulations of protein alone (green), implicit (blue), explicit (orange) simulations (averaged over 20 trajectories), and with Amber all-atom simulation of protein alone (pink) and of protein with C_60_ (purple). For simulations performed with the UNRES force field, the straight line represents the average value, while the shaded area along the line corresponds to its standard deviation.

**Figure 5 molecules-29-01919-f005:**
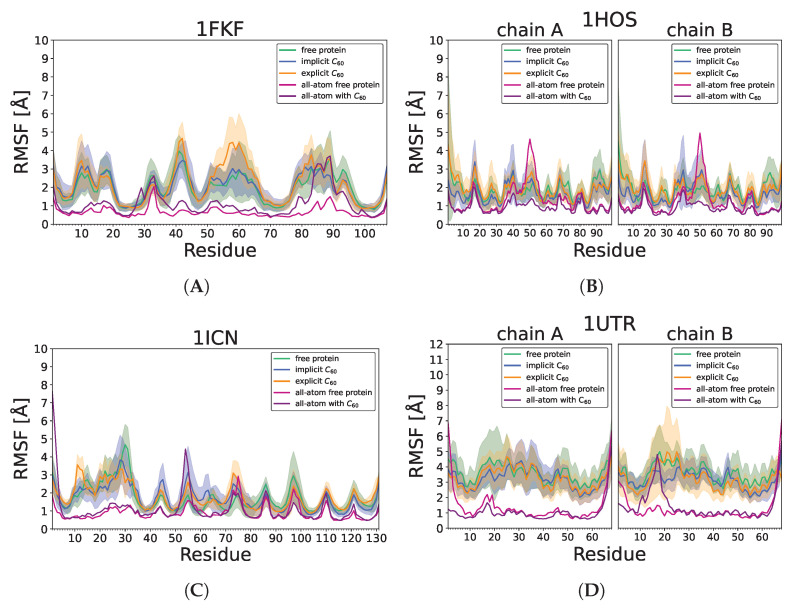
Cα RMSF values in protein-only, implicit and explicit simulations (averaged over 20 trajectories), and during single all-atom simulations in Amber (without and with C_60_) for (**A**) 1FKF, (**B**) 1HOS, (**C**) 1ICN, (**D**) 1UTR, (**E**) 1DPX_no SS_, and (**F**) 1DPX_SS_. For protein-only, implicit, and explicit simulations, the straight line represents the average value, while the shaded area along the line corresponds to its standard deviation.

**Figure 6 molecules-29-01919-f006:**
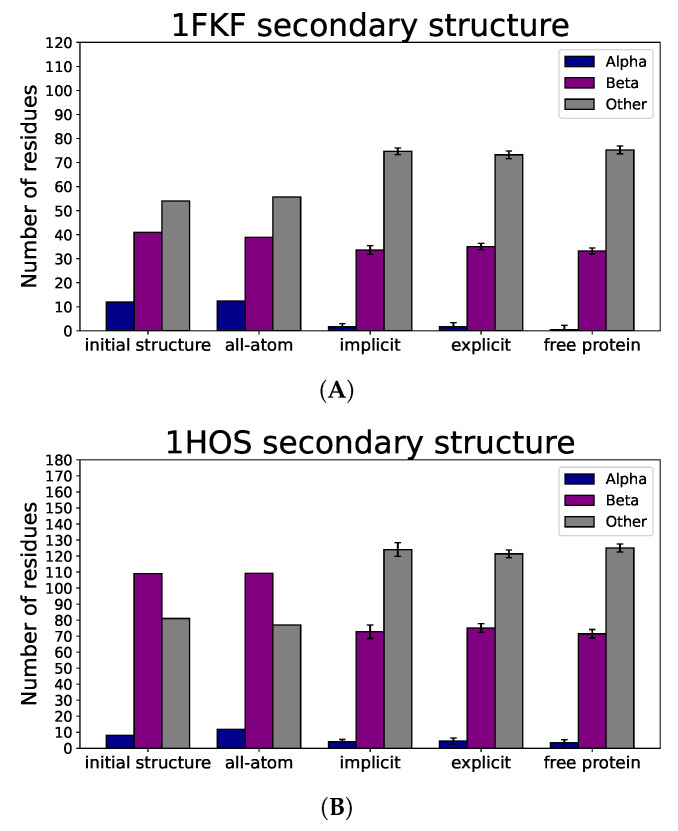
Number of residues (for initial protein structure) and average values of residues (for all-atom simulation with C_60_ in Amber force field, implicit, explicit simulations with C_60_ and protein-only simulations in UNRES coarse-grained force field) in given secondary structure for (**A**) 1FKF, (**B**) 1HOS, (**C**) 1ICN, (**D**) 1HOS, (**E**) 1DPX_no SS_, and (**F**) 1DPX_SS_. Values for UNRES simulations were averaged over 20 trajectories.

**Figure 7 molecules-29-01919-f007:**
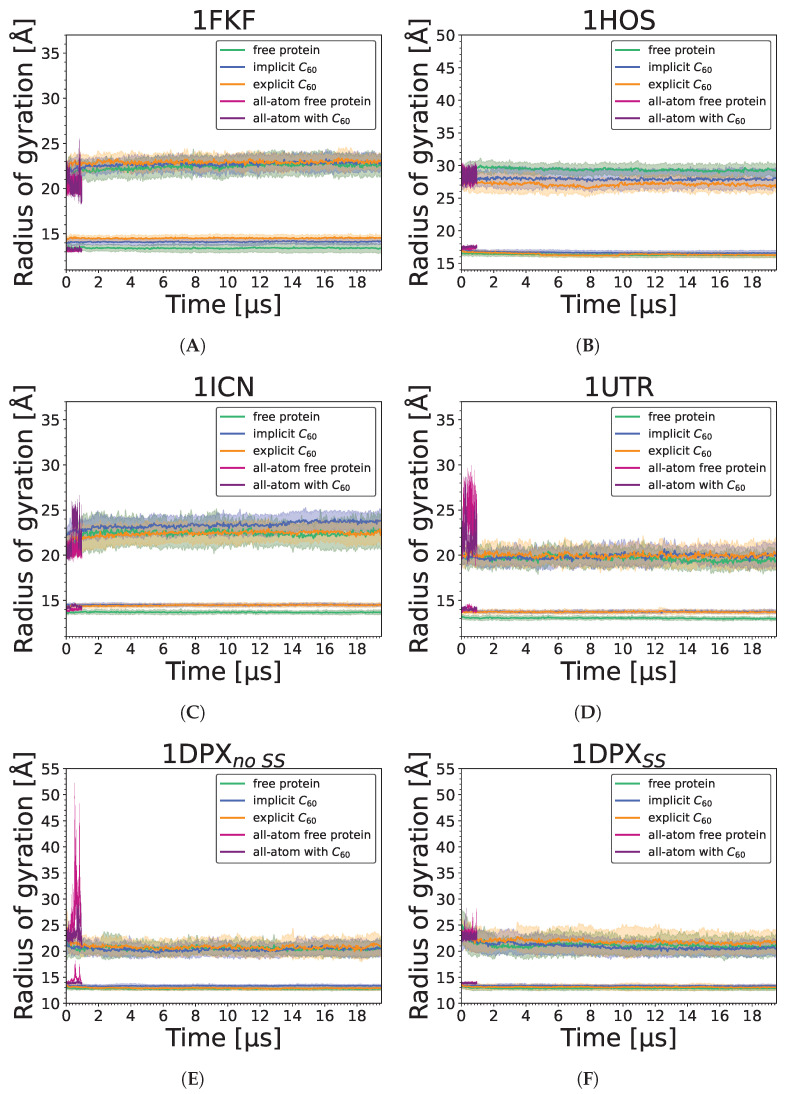
Changes in radius of gyration (bottom lines) and maximum radius of gyration (top lines) during protein-only, implicit and explicit simulations (averaged over 20 trajectories), and during single all-atom simulations in Amber (without and with C_60_) for (**A**) 1FKF, (**B**) 1HOS, (**C**) 1ICN, (**D**) 1UTR, (**E**) 1DPX_no SS_, and (**F**) 1DPX_SS_. For protein-only, implicit, and explicit simulations, the straight line represents the average value, while the shaded area along the line corresponds to its standard deviation.

**Figure 8 molecules-29-01919-f008:**
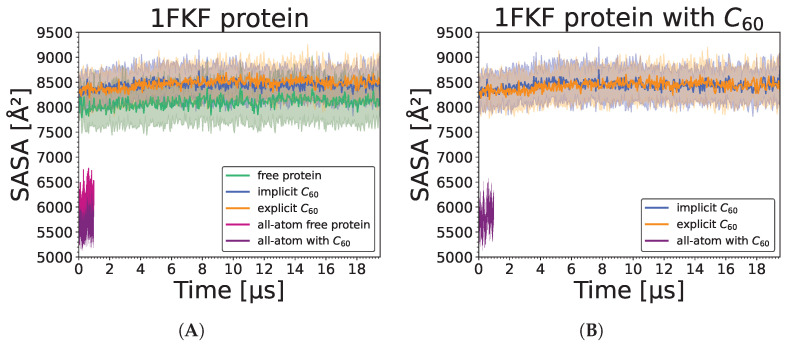
SASA values for protein structures (**A**,**C**,**E**,**G**,**I**,**K**) in protein-only, implicit, explicit simulations (averaged over 20 trajectories), and during single all-atom simulations in Amber (without and with C_60_) and for proteins with C_60_ (**B**,**D**,**F**,**H**,**J**,**L**) in implicit and explicit simulations (averaged over 20 trajectories), and single all-atom simulation in Amber. For protein-only, implicit, and explicit simulations, the straight line represents the average value, while the shaded area along the line corresponds to its standard deviation.

**Figure 9 molecules-29-01919-f009:**
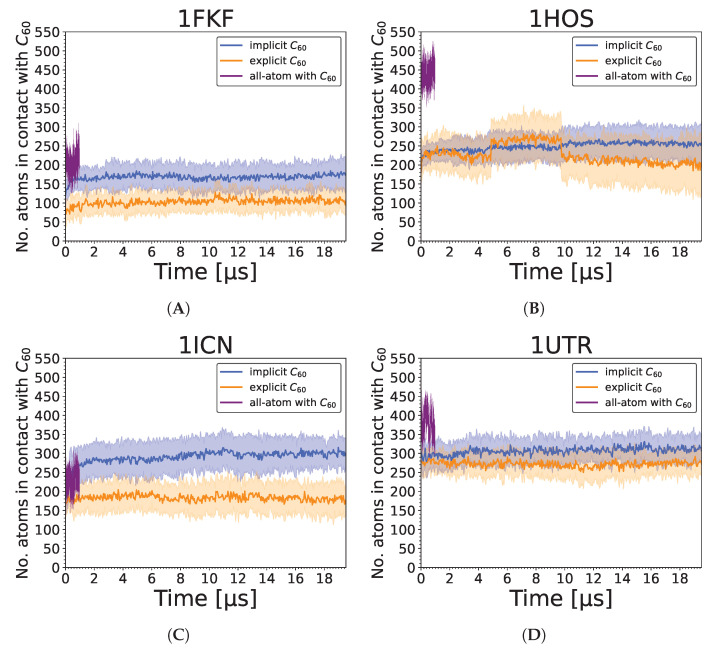
Number of atoms in contact with C_60_ during implicit and explicit simulations (averaged over 20 trajectories), and during single all-atom simulation in Amber for (**A**) 1FKF, (**B**) 1HOS, (**C**) 1ICN, (**D**) 1UTR, (**E**) 1DPX_no SS_, and (**F**) 1DPX_SS_. For implicit and explicit simulations, the straight line represents the average value, while the shaded area along the line corresponds to its standard deviation.

**Figure 10 molecules-29-01919-f010:**
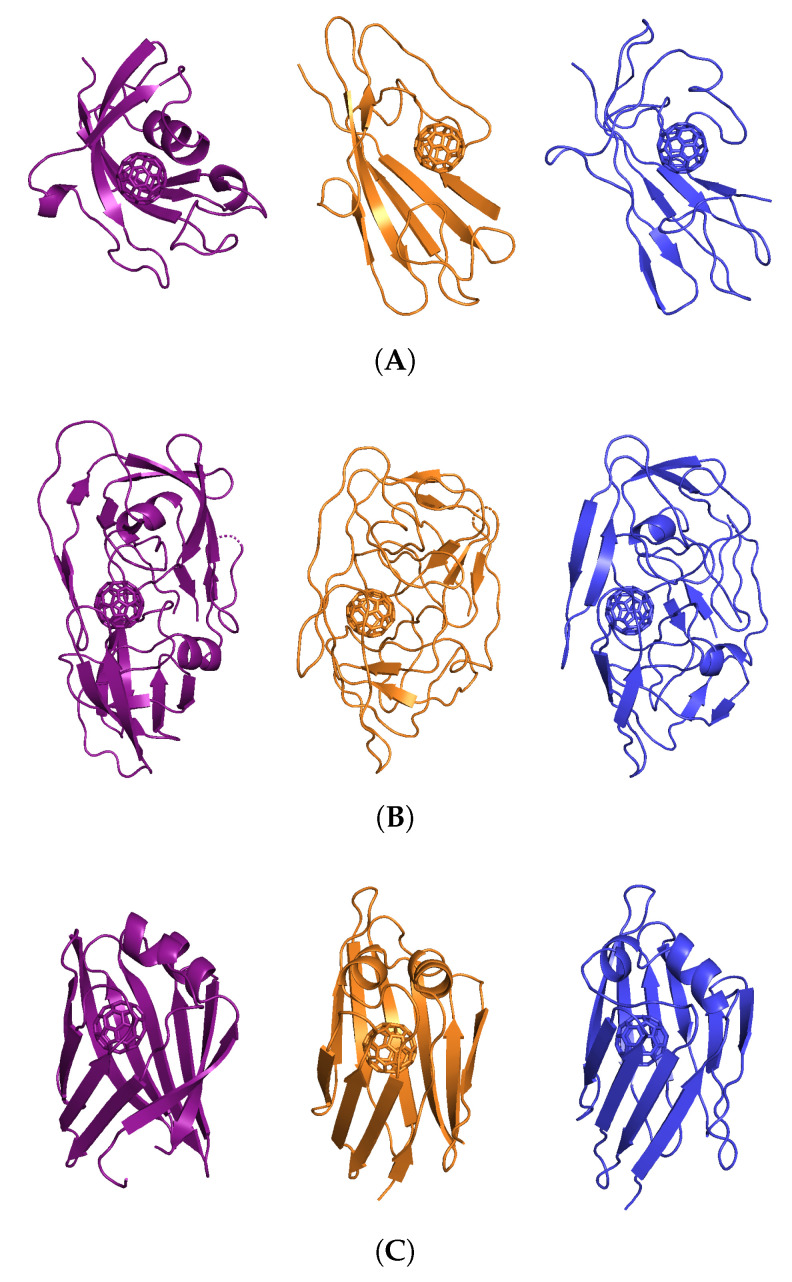
Representative structures of protein with C_60_ from the cluster analysis, where the chosen structure corresponds to the most populated cluster, for (**A**) 1FKF, (**B**) 1HOS, (**C**) 1ICN, (**D**) 1UTR, (**E**) 1DPX_no SS_, and (**F**) 1DPX_SS_. The representatives are shown for all-atom, explicit, and implicit simulations in purple, orange, and blue, respectively.

**Figure 11 molecules-29-01919-f011:**
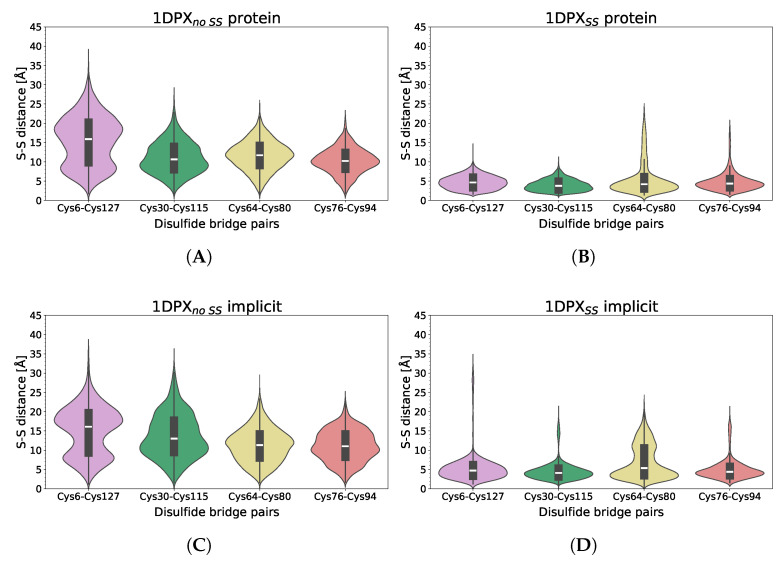
Violin plot of distances between sulfur atoms in UNRES MD simulations of the 1DPX (top row, panels **A**,**B**), with the implicit C_60_ model (middle row, panels **C**,**D**) and with the explicit C_60_ model (bottom row, panels **E**,**F**), with no disulfide bonds (left column, panels **A**,**C**,**E**) and with dynamic treatment of the disulfide bonds (right column, panels **B**,**D**,**F**).

**Table 1 molecules-29-01919-t001:** Average energy of interaction between C_60_ fullerene in implicit and explicit representations in UNRES coarse-grained force field and in all-atom representation in Amber force field (ΔH stands for effective binding energy, while ΔG is Gibbs free energy).

Protein	Energy [kcal/mol]			
	UNRES		Amber	
	Implicit C_60_	Explicit C_60_	ΔH	ΔG
1FKF	−58.29 ± 3.67	−30.33 ± 4.82	−49.02 ± 3.95	−28.55± 5.40
1HOS	−70.58 ± 3.67	−35.28 ± 6.23	−48.66 ± 2.99	−29.40± 6.16
1ICN	−84.48 ± 3.62	−53.96 ± 4.75	−59.83 ± 2.78	−41.36 ± 3.50
1UTR	−90.48 ± 4.15	−61.81 ± 6.22	−70.87 ± 3.81	−50.18 ±.6.27
1DPX_no SS_	−73.00 ± 10.45	−3.46 ± 5.56 *	−24.63 ± 2.15	−8.79 ± 3.67
1DPX_SS_	−60.32 ± 7.85	6.24 ± 5.59 *	−23.26 ± 3.29	−9.06 ± 2.34

* The averages are calculated only for trajectories in which dissociation was not observed.

## Data Availability

Data are contained within the article and Appendix A.

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
