# Peer review of "Integrating Explicit and Implicit Fullerene Models into UNRES Force Field for Protein Interaction Studies"

_molecules, 2024, doi:10.3390/molecules29091919_

Round 1

Reviewer 1 Report

Comments and Suggestions for Authors

The authors report the creation of C60 fullerene models for use in molecular dynamics sessions within the coarse-grained UNRES force field. Both explicit and implicit models are provided, along with a comprehensive list of interaction data collected from multiple MD runs on a set of protein-C60 complexes (proteins selected as the most likely C60 targets, hand-prepared complexes) and compared to the obtained results with the all-atom AMBER force field.    

The article is clearly and well written, examines the strengths and weaknesses of the UNRES methodology and deserves to be published, minor concerns aside, as follows:

1. RMSD and RMSF data using the UNRES force field show much larger protein fluctuations than using the all-atom force field, including changes in the alpha-helix/beta-sheet ratio in secondary structure. More importantly, in many cases the RMSD appears not to reach a plateau after 10 µs. The authors discuss this point in lines 434-444. While not directly related to C60 modelling, it should be important for readers planning to use the UNRES force field to know whether work is underway to overcome this inherent limitation; or perhaps they could try to convince readers that this is not one of the major drawbacks of the method: if the goal is to lengthen the simulation duration by several orders of magnitude, this is critically important.

2. The color coding in the figure and caption is different.

Reviewer 2 Report

Comments and Suggestions for Authors

The manuscript describes the development and incorporation of two models (explicit and implicit) of fullerene C60 into the UNRES coarse-grained force field. My overall impression of the study is very good. I find that the manuscript is overall well-organized and clear. The methodology is adequate and properly described. The results are clearly stated and the conclusions are supported by the obtained results.

I would suggest the authors focus the Introduction section a little bit, as now it is spilling out too much into unnecessary details. I would also recommend that they add the used σ and ε parameters in the implicit model into the methodology section or possibly somewhere in the Supplementary file. Also, why can the “MARTINI coarse-grained force field not be effectively used to study large conformational changes of proteins”? Please, add a relevant reference for this statement.

In conclusion, I recommend that the manuscript be accepted for publication in the journal Molecules after some rather minor revisions.

Reviewer 3 Report

Comments and Suggestions for Authors

In this paper, we successfully integrated explicit and implicit C60 fullerene models into the UNRES coarse-grained force field for molecular dynamics simulations of protein-nanoparticle interactions. By analyzing the interaction patterns between different proteins and C60, specific interaction mechanisms and binding sites were revealed.
However, the article still suffers from the following problems:
1. The abstract section is missing the significance of this study
2. There is insufficient description of the experimental details, and the specific implementation details of the explicit and implicit models are not described in sufficient detail.
3. For the choice of protein type, it should be explained why these proteins were chosen.
4. In terms of theoretical analysis and interpretation of experimental results, the article fails to delve into the possible mechanisms behind the observed phenomena. For example, the differences in the binding sites and binding modes of C60 with different proteins, and how these differences affect the structure and function of the proteins.
5. In line 5 of abstract “The UNRES force field offers computational efficiency, allowing for longer timescale simulations while maintaining accuracy” should be preceded by this sentence in line 3: “This study implements both explicit and implicit C60 models into the UNRES coarse-grained force field, enabling the investigation of fullerene-protein interactions without the need for restraints to stabilize protein structures.

Comments on the Quality of English Language

I am not qualified to evaluate the English

Round 2

Reviewer 3 Report

Comments and Suggestions for Authors

This study can be accepted at this version.

Comments on the Quality of English Language

This study can be accepted at this version.